# The MST1R/RON Tyrosine Kinase in Cancer: Oncogenic Functions and Therapeutic Strategies

**DOI:** 10.3390/cancers14082037

**Published:** 2022-04-18

**Authors:** Alex Cazes, Betzaira G. Childers, Edgar Esparza, Andrew M. Lowy

**Affiliations:** Moores Cancer Center, Department of Surgery, Division of Surgical Oncology, University of California San Diego, La Jolla, CA 92037, USA; alex.cazes.fr@gmail.com (A.C.); childebr@ucmail.uc.edu (B.G.C.); edesparza@health.ucsd.edu (E.E.)

**Keywords:** RON, MST1R, tyrosine kinase, MET, cancer

## Abstract

**Simple Summary:**

MST1R/RON receptor tyrosine kinase is a highly conserved transmembrane protein present on epithelial cells, macrophages, and recently identified in a T-cell subset. RON activation attenuates inflammation in healthy tissue. Interestingly, it is overexpressed in several epithelial neoplasms with increasing levels of expression associated with worse outcomes. Though the mechanisms involved are still under investigation, RON is involved in carcinogenesis via immune modulation of the immune tumor microenvironment, activation of numerous oncogenic pathways, and is protective under cellular stress. Alternatively, inhibition of RON abrogates tumor progression in both animal and human tissue models. Given this, RON is a targetable protein of great interest for cancer treatment. Here, we review RON’s function in tissue inflammation and cancer progression, and review cancer clinical trials to date that have used agents targeting RON signaling.

**Abstract:**

The MST1R/RON receptor tyrosine kinase is a homologue of the more well-known MET receptor. Like MET, RON orchestrates cell signaling pathways that promote oncogenesis and enable cancer cell survival; however, it has a more unique role in the regulation of inflammation. RON was originally described as a transmembrane receptor expressed on tissue resident macrophages and various epithelial cells. RON is overexpressed in a variety of cancers and its activation modifies multiple signaling pathways with resultant changes in epithelial and immune cells which together modulate oncogenic phenotypes. While several RON isoforms have been identified with differences in structure, activation, and pathway regulation, increased RON expression and/or activation is consistently associated with worse outcomes. Tyrosine kinase inhibitors targeting RON have been developed, making RON an actionable therapeutic target.

## 1. Introduction

Récepteur d’Origine Nantais (RON) is a receptor tyrosine kinase (RTK) identified in 1993 by Ronsin et al. as a homologue to cMET and a member of the MET proto-oncogene family [1]. The RON protein is encoded by the *MST1R* gene, located on chromosome 3 in humans, and is highly conserved across species. RON is translated as a single transmembrane pro-protein which is subsequently cleaved in its extracellular portion by proteases. A 40 kDa-α chain, solely extracellular, is released and binds to the remaining transmembrane beta chain of 150 kDa. The subsequent receptor is composed of an extracellular sema domain, a transmembrane domain, and an intracellular portion containing the kinase domain. RON is activated by the binding of the macrophage-stimulating protein (MSP), also known as macrophage stimulating 1 (MST1), the sole ligand to RON that has been identified to date [2,3]. MSP is expressed as a pro-protein in the liver, lungs, adrenal glands, placenta, kidneys, and pancreas and requires a proteolytic cleavage to become active [4].

RON may be activated by (1) ligand binding, (2) homodimerization, (3) heterodimerization to other tyrosine kinases, (4) constitutive activation as an alternative isoform or splice variant, and (5) in rare instances, activating point mutations have been identified (Figure 1). Upon activation, RON receptor activation triggers a downstream signaling cascade that ultimately results in the activation of multiple molecules and pathways, including β-catenin/TCF-4, Src, Ras/Raf/MEK/ERK/MAPK, JNK/STAT, SMAD/TGFβ, and PI3K/AKT [5,6,7,8]. RON is also involved in the development of the epithelium, brain, and neuroendocrine tissues [9]. In the adult, RON is expressed in the epithelia of the liver, lung, gut, kidney, bone, adrenal gland, and skin [10]. RON is also expressed on the surface of macrophages and CD34^+^ normal hematopoietic stem cells [11,12]. In non-carcinogenic tissue, RON is involved in attenuating an inflammatory response. The effects of insufficient or excess RON on inflammation are further detailed in the Section 2 below.

In addition, aberrant activation of the receptor has been described in many solid tumors. These include pancreas, lung, liver, breast, colon, prostate, bladder, and ovarian cancers, as well as AML and Burkitt lymphoma [13,14,15,16,17,18,19,20,21,22,23,24,25]. RON activation contributes to tumor progression and metastasis by promoting cell proliferation, motility, and inhibiting apoptosis. In the tumor microenvironment (TME), RON is expressed in epithelial tumor cells, tumor-associated macrophages (TAMs), tumor-associated myeloid-derived suppressor cells (MDSCs), and cancer-associated fibroblasts (CAFs) [26]. Overall, RON activity in cancer is the result of a complex cascade of events induced by RON activation leading to direct effects on tumor progression on epithelial cancer cells and an indirect effect through the modification of immune phenotypes in the tumor microenvironment toward one that is tumor-permissive. Evaluation of the role of RON in each individual cell type is needed to better understand the mechanisms by which RON regulates tumorigenesis.

## 2. RON Attenuates Inflammation

Various disease models have demonstrated that under normal biological conditions, RON attenuates inflammation by decreasing the secretion of pro-inflammatory cytokines. These mechanisms are important to understand as alterations in the immune microenvironment can affect tumor progression. RON activation in macrophages occurs following MSP binding. MSP is secreted as an inactive form and requires activation by serine proteases including matripase, hepsin, and HGF-A. Mice harboring a kinase dead mutation RON designated as (TK-/-) have an increased susceptibility to nickel-induced acute lung injury with clusters of cells in lungs producing granzymes and composed of macrophages, lymphocytes, and neutrophils [27]. In a contrasting model, MSP activation of RON led to a decrease in the production of TNF-α by alveolar macrophages following and during LPS stimulation. This led to a less severe form of acute lung injury with significant alveolar wall thickening and protein leakage [28]. Activation of RON inhibits LPS-induced degradation of Iκβ-α, thus inhibiting NF-κβ nuclear translocation [29]. This in turn leads to a reduction in shock mediators, prostaglandin E2 (PGE2) and COX-2 inducible enzyme [29]. In murine alveolar macrophages, RON activation leads to a decrease in NF-κβ activity. NF-κβ is responsible for regulating TNF-α at the transcriptional level [29]. In addition, the enzyme ADAM17 cleaves TNF-α protein for activation and is found to be downregulated upon MSP binding of RON and increasingly expressed in RON knockdown macrophage lines [30].

In addition to its modulation of shock mediators, RON activation induces gene expression patterns characteristics of anti-inflammatory macrophages such as the induction of Arginase I and the suppression of the pro-inflammatory marker iNOS. This combined effect leads to the conversion of arginine into ornithine in favor of nitric oxide, a free radical and a major resource of oxidative stress. Effects on Arginase I expression are observed only when macrophages are stimulated with LPS, indicating that RON modifies polarization following TLR-4 activation but does not induce an anti-inflammatory state on its own. RON signaling also suppresses the production of pro-inflammatory cytokines after LPS or IFN-γ stimulation [22]. These notions are revisited in the section reviewing macrophage modulation in cancer. The importance of RON regulated inflammation is depicted in a simian immunodeficiency model identifying an inverse relationship between levels of RON expression and damage to the central nervous system secondary to inflammation. Real-time RT-PCR demonstrated a 60%< reduction in RON expression in the brains of animals with CNS lesions compared to those of uninfected controls. Disease progression was also associated with an increase in inflammatory cytokine TNF-α and a decrease in immune suppressive Arginase I from tissue samples at progressive time intervals. Not surprisingly, an inverse relationship between viral load and RON was also described [31]. RON is involved in downregulating the damaging tissue-specific effects of unchecked inflammation under normal homeostatic conditions. Overall, increasing levels of RON activity are associated with decreased inflammation, whereas decreasing levels of RON activity promote an inflammatory state.

## 3. RON Isoforms

In the last decade, several functional isoforms of RON have been identified, primarily resulting from post-translational splicing alterations of full-length RON. Several of these are specific to certain tumor types. Some RON isotypes are constitutively active. Other isoforms lack the extracellular domain requiring new drug binding strategies to inhibit kinase activity.

Known isoforms include 1254T, 170, 160E2/E3, 160, P5P6, 155, 110, 85, and 55 (sfSMT1R) [26,27,28]. A subset of isoforms, 140, 155, 160, 165, P5P6, and sfRON are known to exist in a constitutively phosphorylated state with several originally identified in human colorectal adenocarcinoma [23]. Constitutively active RON isoforms have demonstrated up to 90% inhibition of LPS-induced COX-2 protein and mRNA expression even in the absence of MSP [13]. Furthermore, both MSP dependent and MSP independent isoforms 155, 160, and 165 induced scatter phenotypes after plating when transfected into Madin Darby canine kidney (MDCK) cells. These also demonstrated transformative cell properties and anchorage-independent growth in transfected NIH3T3 cells in both in vitro and in vivo colorectal cancer models [29]. In 2013, Moon et al. utilized mutagenesis analysis and stepwise base substitutions to identify enhancers of exon 11 inclusion of RON pre-mRNA and pinpointed the 2-nt RNA of their exon 11 mutant, 11-3. In addition to the wild-type AG sequence, nucleotide pairs GA, CC, UG, and AC enhance inclusion of exon 11, while base mutants UA, GC, UU, and GG obliterate inclusion of exon 11 [32].

Chadekis et al. characterized RON isoform expression in pancreatic cancer cell lines and patient-derived pancreatic cancer xenografts. An increase in isoforms 165, P5P6, and sfRON was noted as the overall RON expression increased. Isoform 165 represented up to 30% of the total RON transcript in high RON expressing xenografts. In addition, these three isoforms constituted 42% of the total RON transcript in the PDX lines evaluated. They were the first to report that both P5P6 and sfRON have in vivo transformative tumorigenic activity in pancreatic cancer models [33,34]. The transcriptome associated with the expression of these variants differed between isoforms, demonstrating the need for additional studies to better understand isoform function.

In 2011, sfRON was identified as the predominant phosphorylated RON isoform in primary human breast cancer samples. sfRON was expressed in the breast cancer line MCF7 and associated with PI3K pathway activation leading to increased migratory capacities in vitro and larger primary orthotopic tumors in vivo. Differences in pathway activation were noted between sfRON and wtRON with the former signaling via PI3K activation and MAPK inhibition whereas both pathways were active with wtRON [29].

Recently, Lai et al. indicated for the first time that sfRON is expressed in T-cells and inhibits Th1 differentiation of CD4^+^ immature T-cells leading to blunting of an anti-tumor response. Furthermore, sfRON can attenuate recruitment and trafficking of anti-tumor T-cells from lymph nodes to the TME. Knockout of sfRON essentially obliterated establishment and propagation of metastatic lesions in this breast cancer mouse model [35]. Furthermore, transfer of sfRON knockout CD4^+^ T-cells to RON WT mice was largely protective against metastatic outgrowth following tumor cell injections.

In addition to breast and pancreatic cancer, sfRON is known to be expressed in gastric cancer. Wang et al. demonstrated enhanced cancer cell proliferation via significantly upregulated glucose metabolism intermediates in high sfRON expressing gastric cancer human tissue samples compared to RON overexpressing samples or control samples [36]. In addition, gene set enrichment analysis and qRT-PCR analysis confirmed the upregulation of glucose metabolism intermediates in GTL-16 and MKN-45 gastric cancer cell lines. Further investigation identified *SIX1* as the effector of the sfRON/β-catenin pathway, which leads to tumor growth, and enhancement of glycolytic genes *GLUT1*, *LDHA*, and *HK2* (Figure 2). Greenbaum et al. transfected low RON expressing HEK293 cells with wtRON, isoforms 155, 160, and 165 and found increased motility via scratch assay in all lines relative to control. Again, gene expression analysis of RON expressing cell lines demonstrated isoform specific variation [37]. Differences in downstream signaling induced by RON isoforms has also been shown to differ by cancer type [27,28,29,30,31,32,33,34,35,36,37,38,39,40].

Characterizing RON kinase expression in lung cancer has led to the identification of novel isoforms, including the deletion of exon 18 and 19 in the C-terminus region of RON [31]. Krishnaswamy et al. evaluated numerous SCLC and NSCLC cell lines using cDNA, exon specific primers, and PCA amplification to identify four novel RON isoforms. These included the skipping of exons 15–19, 16–19, 16–17, and 16 alone [41]. These transcript variants consist of exon skipping within the kinase domain. This specific study did not evaluate isoform function. Clearly, it is important to understand both the mechanisms underlying the expression of the many isoforms of RON in addition to the biological impact of different isoforms in the context of specific cancer types in order to generate rational strategies for designing therapies.

## 4. RON Alters Macrophage Polarization

Macrophages possess broad modulatory and effector repertoires by which they serve two overarching purposes. The first is to protect the host by upregulating production of pro-inflammatory cytokines, reactive oxygen intermediates, tumoricidal activity, and promotion of Th1 cells. Macrophages are also responsible for resolving inflammation and aiding in tissue rebuilding and remodeling following an insult [16]. Macrophages can exist anywhere along this pro- or anti-inflammatory continuum with LPS and IFN-γ being potent stimulators of inflammatory cascades while IL-4/IL-13 induce pathways of immune suppression [17]. Their fate is influenced by signals received during maturation, activation, and immune engagement in the form of molecular, epigenetic, or host specific signaling [17,18]. The dichotomous classification of these largely in vitro-derived extremes refers to them as classical M1 macrophages, known for their pro-inflammatory and anti-tumorigenic properties, whereas alternative M2 macrophages are immunosuppressive with pro-tumorigenic qualities (Figure 3) [16,19]. RON alters macrophage polarization with implications for cancer biology. RON activation suppresses the anti-tumorigenic M1 macrophage phenotype by inhibiting STAT1 phosphorylation and NF-kB activation induced by IFN-γ and LPS, respectively [20]. Several studies in mice continued to expand our understanding of the tumor-specific changes that occur as well as their implications in tumor regulation. Transcriptional profiling in FVB mice, with M2-biased peritoneal macrophages, demonstrated that MSP stimulation of intact RON can lead to significant downregulation of genes in the IFN-γ pathway and significant upregulation of genes involved in immune tolerance and tissue repair [18]. The IFN-γ pathway is known to have inhibitory effects on tumor initiation and promotion by modulation of innate and adaptive immunity [21]. Increased expression of RON correlates with increased expression of Arginase I, a pro-tumorigenic enzyme characteristic of M2 macrophages [19]. Activation of RON in the tumor microenvironment may facilitate tumor survival by hijacking inflammatory and tissue repair pathways to promote self-survival against host defenses [22]. FVB mice tumor models of papilloma, fibrosarcoma-FVB, and methylcholanthrene demonstrated slowed tumor initiation, and overall outgrowth with decreasing levels of RON. In another mouse model, Gurusamy et al. demonstrated a significant reduction in tumor size and tumor cell apoptosis in tumors from (TK-/-) hosts after transgenic TRAMP-C2R33 prostate cancer cell line orthotopic injections. In addition to altering macrophage phenotype, RON expression results in the modification of macrophage migration ability. Many researchers have noted a decrease in F4/80^+^ CD68^+^ macrophages within the TME of RON knockdown models across various cancer types [1]. However, Gurusamy described increased intratumoral macrophage infiltration in RON TK-/- derived tumors. Subsequent in vitro analysis of macrophage migration using the immortalized murine alveolar macrophage line MH-S with RON expression (shNT) or RON knockdown (shRON) revealed significant increases in migration ability in the RON knockdown (sHRON) macrophage cohort. It is consistent that RON function impacts macrophage migration; however, additional work is needed to pinpoint how RON specifically impacts macrophage populations across cancer types. Increased RON macrophage expression is noted to alter cell signaling pathways suppressing CD8^+^ T-cell activation associated with cancer progression (Figure 2) [2,3,4]. CD8^+^ T-cell mouse depletion studies negated the benefits of RON knockdown, implying interplay between RON expressing macrophages and T-cell regulation for tumor control. Only recently has RON expression been demonstrated in T-cells with an associated blunting of anti-tumor response [35]. In TK-/- mouse models with increased F4/80 macrophage infiltration, Annexin V/PI staining demonstrated increased cell death compared to WT. Together, these findings highlight the importance and influence host RON status can have on tumor growth, macrophage modification, and immune modifying T-cell interactions [9].

## 5. RON in Cancer Cells

In humans, RON is overexpressed in up to 50% of breast cancers, 40% of colorectal cancers, over 80% of human pancreatic cancers, and 90% of prostate cancers. Overexpression promotes tumor growth in these and other cancers [2,10,11,12,13,14]. Furthermore, it is well-documented that RON is minimally expressed in benign tissue types and increasingly expressed with cancer progression and is typically maintained in metastases [14,15,16,17]. This is clinically significant since increased RON expression has been associated with worse clinical prognosis in breast, colorectal, prostate, and pancreatic cancer [14,15,17,18]. Here, we review what is known about specific human cancers as well as the ongoing investigation in respective animal models.

Welm et al. assessed microarray gene data of 295 breast cancer patients from the Netherlands Cancer Institute and noted decreased time to metastasis and decreased overall survival in patients with concomitant overexpression of RON, MSP, and MT-SP1. Furthermore, concomitant overexpression of these genes was an independent predictor for poor outcome and when considered in combination with a 70 gene prognostic signature, was more accurate in predicting five-year metastasis than either alone [13,38].

Animal models evaluating metastasis between RON WT and Ron TK-/- hosts found a significantly decreased lung tumor burden in RON TK-/- hosts leading to an improvement in overall survival by 50% compared to RON WT hosts. A defect in supporting the conversion of seeded metastasis to overt metastatic colonies was reproduced across several cell lines including polyomavirus middle T antigen (PyMT-MSP), polyomavirus MSCV-IRES-GFP (PyMT-MIG) control cells, lung alveolar/bronchiole carcinoma-P0297 (LAP-MSP), and lung alveolar/bronchiole carcinoma-P0297 control (LAP-MIG) [13,21,38,39].

Consistent with these findings, expression of RON increases during the progression of colorectal cancer and is associated with worsened tumor differentiation [14,42]. In pancreatic cancer, human tissue samples demonstrate progressively higher levels of RON expression during progression from pancreatic intraepithelial neoplasia (PanIN) to pancreatic adenocarcinoma [17,33,34,40]. Furthermore, analysis of the human TGCA pancreatic cohort revealed an association between RON expression and decreased disease-free survival and overall survival [17,40]. Given that KRAS is mutated in up to 90% of pancreatic cancers, mouse models with the same mutation have been used to study the role of RON [18,33,34,40]. Babicky et al. demonstrated that RON overexpression in mice expressing oncogenic KRAS developed more rapid progression of pancreas-specific acinar-ductal metaplasia and PanIN lesions compared to age-matched KRAS only mutated mice. In addition, overall survival was strikingly reduced in KRAS^LSL-G12D^/RON/Cre (KRC) mice compared to KRAS^LSL-G12D^/Cre (KC) mice. These studies demonstrated that RON promotes both tumor initiation and progression in KRAS-driven pancreatic cancer. Further supporting this are findings that KRAS^LSL-G12D^/RON TK-/- mice have slower pancreatic cancer onset, progression, and prolonged survival when compared to KC models with physiologic levels of RON [40]. RON is also overexpressed in hepatocellular carcinoma [22]. In this cancer type, expression of cytokines IL-6, TNF-α, IL1-α, and HGF are associated with increasing levels of RON expression [22,43]. RON’s expression across numerous solid tumors as well as its consistent association with worse outcomes would make effective therapeutic strategies applicable across many cancer types.

## 6. RON Crosstalk and Other RTKs

In addition to ligand binding and homodimerization, receptor tyrosine kinase (RTKs) can be activated by heterodimerization. These physical and functional interactions have been shown to play a role in tumor progression and can contribute to treatment failure. Crosstalk between RON and several other proteins has been described and is implicated in tumorigenesis in several solid tumors.

### 6.1. RON and EGFR Crosstalk in Cancer

Co-expression of RON and EGFR has been reported in several tumor models including lung, colorectal, liver, and breast. Cooperation between RON and EGFR has been previously reported in bladder cancer as they are co-expressed in one third of patients [25]. This co-expression is associated with tumor invasion and decreased survival. In vitro inhibition of EGFR modulates RON activity in bladder cancer cell lines demonstrating the interplay between these two RTKs [44,45]. RON has also been implicated in promoting tumor growth in head and neck squamous cell carcinomas (HNSCCs) [46]. In 2013, Keller at al. observed RON expression in 64% of primary tumors. This expression was associated with both EGFR expression (*p* < 0.01) and EGFR activation (*p* < 0.001) [46]. In vitro experiments revealed that RON interacts and synergizes with EGFR to promote cell migration and proliferation. The authors showed that RON and EGFR can transactivate when stimulated by their respective ligand [46]. Another interaction between RON and EGFR occurs in protein complexes that also contain syndecans and integrins at the cell surface. Interaction with EGFR results in the stabilization of this EGFR–RON complex during times of cellular stress, possibly resulting in the prevention of cell cycle arrest via c-Abl in both HNSCC and breast carcinomas [47]. Such control of the cell cycle does not seem to depend on EGFR activity and could explain the high number of tumors refractory to EGFR inhibition.

In colorectal cancer, RON and cMET RTKs were activated as a result of treatment with cetuximab, an anti-EGFR monoclonal antibody. This activation conferred treatment resistance [42]. In this publication, Graves-Deal et al. also showed that crizotinib, an inhibitor of cMET and RON, was able to circumvent cetuximab-acquired resistance. This overlap between RTK functions indicates that in CRC, as in many other solid tumors, a broad spectrum of kinase inhibition may be more effective than single target approaches [42].

### 6.2. RON and MET Crosstalk in Cancer

RON and MET receptor tyrosine kinases belong to the same subfamily and are 68% homologous. Both receptors can activate signaling pathways such as PI3K/AKT and MEK/ERK and have been implicated in tumorigenesis through the regulation of cell proliferation, apoptosis, metastasis, angiogenesis, maintenance of cancer stem cells, and resistance to chemotherapy cells [48,49,50,51,52,53,54]. RON and MET are often co-expressed in cancer and crosstalk between the two receptors has been demonstrated [52,53,54,55,56,57]. Both receptors can form homo- and heterodimers and engage in transphosphorylation [52,53,54,55,56,57]. RON and MET can be targeted by small molecules, with similar IC50′s for certain drugs. More studies and clinical trials have been conducted with MET. While targeting tumors with MET overexpression is possible, the efficacy may be blunted due to functional redundancy, most notably with RON. Here, we reviewed several studies that were conducted to evaluate crosstalk between RON and MET.

It has been reported that in cancers ‘addicted’ to MET signaling, RON phosphorylation is dependent on the level of expression and activation of MET [52]. Benvenuti et al. showed that RON can be transphosphorylated by MET in gastric and lung cancer cell lines. They also demonstrated that RON activation is sensitive to MET-specific molecular inhibitors and that RON knockdown in MET-addicted tumors affects cell proliferation and tumorigenicity [52]. In prostate cancer, co-expression of RON and MET promotes metastasis though ERK1/2 pathway activation. Targeting them using siRNA or the small molecule inhibitor foretinib suppressed in vitro migration and invasion of prostate cancer cells. This suggests that both receptors are necessary to achieve the full metastatic potential of prostate cancer cells [53,54,55,58].

Similarly, co-overexpression of both receptors was reported in triple-negative breast cancer (TNBC) [56]. Weng et al., using in vitro testing of different kinase inhibitors, demonstrated that targeting both MET and RON reduced cell migration, proliferation, and tumor size in vivo using murine xenografts. Despite the lack of validation by genetic approaches, they showed that targeting both RON and MET can have a more potent effect than targeting RON alone [56]. These results suggest that in TNBC, RON and MET heterodimers could more efficiently activate signaling pathways including MEK/ERK or PI3K/AKT [52,53,54,55,56,58]. It is also possible that RON and MET have partially overlapping functions, though the entire spectrum of activity for both receptors is required to reach the maximum effects. Consequently, targeting both receptors might be necessary to achieve a robust anti-tumor response.

The case of pancreatic cancer is of interest as the relative importance of one RTK on the other is controversial. Hu et al. tested several TKI’s on pancreatic cancer cell lines in vitro and in vivo. They observed that TKIs targeting both RON and MET led to a reduction in cell proliferation and migration, whereas specific inhibition of MET had no effect [57]. Interestingly, they observed a similar reduction of cell proliferation and migration using Tivantinib: a MET specific agent that binds dephosphorylated MET kinase rather than the kinase domain. This result suggests a possible role for MET that is independent of its kinase activity. RON also has biological functions independent of its kinase activity which have yet to be fully understood. This is evidenced by the fact that RON constitutive knockout mice are embryonic lethal due to a deficit in peri-implantation, while mice harboring a RON kinase-dead mutation are viable [59]. These kinase-independent functions of RON and MET remain to be characterized. It would be interesting to define if their respective kinase-independent activities harbor functional redundancy.

Another study of RON and MET in pancreatic cancer reached different conclusions. Vanderwerff et al. conducted an evaluation of transcriptomic signatures following MSP and HGF in vitro stimulation of BxPC3 cells [60]. They observed that both ligands led to enhanced migration and activation of ERK and MAPK pathways. Importantly, they also showed that MSP stimulation led to transcriptomic effects like HGF stimulation but recapitulating only a fraction of effects observed after HGF stimulation. The authors concluded that targeting both receptors might be necessary in pancreatic tumors co-expressing the two receptors. One limitation of this work was that only one cell line was investigated, BxPC3. This cell line is poorly representative of the human disease since it does not carry a KRAS mutation which is present in >90% of human pancreatic cancers. RON has been shown to be a key regulator of KRAS mutant phenotypes [16]; therefore, we can speculate that the relatively small contribution of MSP–RON observed compared to HGF–MET may be explained by the fact that the KRAS–RON axis is not the main oncogenic driver in this cell line. It would be ideal to repeat such transcriptomic analyses on additional cell lines.

Finally, MET is expressed by macrophages and stimulates an anti-inflammatory response by modulating cytokine expression [36,57]. In T-cells and B-cells, MET is expressed consequently to TCR and BCR signaling and favors activation and antibody production. In neutrophils, MET is present in granules and is released upon stimulation. In DCs, MET expression renders cells tolerant to the immune reaction. The RON kinase is also expressed in immune cells and regulates the inflammatory response (see RON and immune regulation section). RON and MET have both been described as regulators of macrophages promoting an anti-inflammatory state, and we can then speculate that inhibiting only one receptor would not be sufficient to reverse macrophage polarization if a functional redundancy also exists in macrophages. Further work is required to evaluate the level of crosstalk between RON and MET in immune cells.

### 6.3. RON Crosstalk with Other RTKs

RON crosstalk has most often been reported to occur with MET or EGFR as discussed above. However, Batth et al. reported crosstalk with an androgen receptor (AR) in prostate cancer [55,58]. Androgen deprivation is an initially effective treatment strategy in localized androgen sensitive prostate cancer, though once refractory, it progresses to metastasis. One mechanism of resistance appears to be the reactivation of androgen receptor signaling controlled by RTKs. RON is highly expressed in castrate-resistant prostate cancer and is activated following androgen deprivation to compensate for the loss of AR expression [55,58]. RON activates the transcription of the anti-apoptotic AR target gene c-FLIP by binding to its promoter region. c-FLIP is not the only AR target gene of importance in prostate cancer, and it will be interesting to evaluate the global impact of RON on AR targets and signaling. These findings suggest that inhibition of RON combined with AR antagonists may be an effective therapeutic approach in advanced prostate cancers, a hypothesis which deserves further evaluation. RON transactivation with PDGFR has been reported in human mesangial cells [58]. Physical interaction with PDGFR receptors allows a ligand-independent activation of RON leading to an anti-apoptotic function. Although such RON activation has been demonstrated in IgA nephropathy and not in solid tumors, it shows that such potential crosstalk is possible and could happen in some of the many PDGFR-altered tumors.

In pancreatic cancer, RON has been shown to interact with insulin-like growth factor-1 (IGFR-1) and becomes activated by IGF1-R after IGF1 stimulation [61,62]. This activation modifies IGF1-R-associated transcriptomic signatures and promotes migration induced by IGF1 stimulation. Another study linked RON and IGF1-R, revealing that RON is expressed in rhabdomyosarcomas and Ewing tumors, both childhood sarcomas [62]. The authors identified RON as a key player enabling resistance to IFG1-R inhibition, by serving as an alternative activator of IGF1-R signaling molecules. Finally, Conrotto et al. described the interaction between RON and plexins which are cell membrane receptors for semaphorins [63]. The authors showed that Semaphorin 4D can indirectly activate RON when interacting with Plexin B1 and that this activation promotes an invasive growth response in vitro. Crosstalk between RON and other kinases contributes to tumor progression and, potentially, to treatment failure in various cancer types by way of redundant pathway activation and overlapping functions. Ongoing research is necessary to delineate these relationships to better guide treatment strategies.

## 7. RON in Metastasis

RON was initially found to regulate cellular motility in macrophages. RON stimulation by MSP induces macrophage cell spreading and attachment in culture as well as chemotactic migration [5]. In addition to cell proliferation and apoptosis, RON is known to regulate cell adhesion and motility of cancerous epithelial cells, notably through integrin-related attachment to extracellular matrix (ECM) [64,65]. Epithelial to mesenchymal transition (EMT) is an essential step in the process of metastasis but also takes place during kidney fibrosis as part of the progression of chronic kidney disease (CKD). In addition to promoting the expression of RTKs involved in CKD, such as VEGFR, PDGFR, and IGFR, it was shown that RON is able to promote EMT in normal kidneys and leads to the expression of fibrotic markers, such as N-cadherin, vimentin, and TGFβ [59]. Several studies have sought to decipher the molecular mechanisms by which RON regulates metastasis. In breast cancer, immunostaining of human primary tumors and paired metastases confirmed the association of RON expression within metastatic deposits [64,65]. In a mouse model, orthotopic implantation of breast cancer MMTV-PyMT-derived cancer cells expressing MSP gives rise to metastatic lesions in the lymph nodes, lungs, spleen, and bones [38]. Interestingly another study demonstrated that lung metastases are absent when the cells are injected in a TK-/- host [39]. Eyob et al. further demonstrated that host RON activity is essential for the transition from micro- to macro-metastasis and acts through the suppression of an anti-tumor CD8^+^ response [39].

Cunha et al. described another mechanism for RON regulation of metastasis. This study conducted on breast cancer xenografts showed that RON promotes metastasis by upregulating the thymidine DNA glycosylase MBD4 [49]. The resulting aberrant DNA methylation profile can be reversed by MBD4 knockdown which in turn blocks metastasis. Similarly, RON inhibitor OSI-296 reverses the methylation profile of genes of the RON/MBD4 epigenetic signature and inhibits lung and lymph node metastasis of patient-derived xenografts [66,67,68,69]. Another study conducted on breast cancer cell lines indicated that RON signaling though the PI3K/mTORC1 pathway promotes metastasis [67,69]. Alternatively, inhibition of both mTORC1 and RON delays the progression of metastases [66,67,68,69]. In the ‘TRAMP’ transgenic mouse model of prostate cancer, the constitutive abrogation of RON kinase activity (TK-/-) or its abrogation in the epithelial compartment completely abolished lung metastasis [15,48].

In gastric cancer, RON seems to mediate metastatic potential via upregulation of UPAR [70] while truncated protein variants of RON seem to play a role in metastasis as well [14,32,33,34,35,36,37,38,39,40,41]. Brain metastasis in patients with solid tumors is associated with a particularly poor prognosis. Two RON mutations located in the tyrosine kinase domain were described in brain metastases from primary lung cancer [71]. A gene polymorphism, previously described in a gastro-esophageal tumor, was found in brain metastases from lung, breast, melanoma, and ovary primary tumors, indicating that RON may play a role in the dissemination to the brain of many cancers. RON’s biological activity in the brain has been reported to modulate regeneration and plasticity by suppressing NO production and acting as a neurotrophic factor [71,72].

In addition, mutations of the MET gene, within the same RON RTK family, were also found in brain metastases and correlated with resistance to radiation therapy in lung cancer [72]. Better characterizing the role of RON in brain-specific metastasis and its effect on the local immune microenvironment could lead to new treatment opportunities in this patient population. CXCR4-CXCL12 has been shown to play a role in tumor growth and metastasis. CXR4 is expressed at the surface of tumor cells and is activated by CXCL12 which is expressed in the stroma or in organs that are preferred sites of metastasis such as lung or bone-marrow. In Ewing’s sarcoma, the CXCR4 antagonist Plerixafor induced cell migration and proliferation by leading to the activation of several RTKs, including RON [73]. The authors described this unexpected result as a compensatory mechanism to sustain cell survival and migratory capacities [73]. Interestingly, CXCR4 was found to be expressed at a higher level in pancreatic cancer cell lines derived from metastatic lesions compared to cell lines derived from primary tumors [74]. Moreover, CXCL12 stimulation of CXCR4 expressing cells promoted cell proliferation and migratory capacities. Plerixafor treatment of a high CXCL12 expressor inhibited proliferation but only partially [74]. The authors did not look for RON expression or activation status in the studied cell lines, but we may speculate that a dual RON/CXCR4 inhibition might improve metastatic spreading in pancreatic cancer patients. RON is part of an elaborate network of kinases and proteins involved in the metastatic process. RON’s role in pathway regulation, tumor cell activity, and disease progression appear to vary based on cancer type. Increasingly, research supports the role of immune cells in regulating metastasis. Notably, the crosstalk between tumor cells and macrophages influences intravasation and immune evasion which are critical steps in the metastatic process. Given RON’s ability to directly and indirectly regulate macrophage function, it is an interesting target to prevent or control metastasis in patients.

## 8. RON and Adaptation to Cellular Stress

Nuclear subcellular localization of receptor tyrosine kinases has been previously reported [74,75,76,77]. Several pertain to nuclear localization of RON kinase and its function in the adaptation to cellular stress. RON can bind to consensus sequences of the genome and behaves as a transcription factor [75]. In 2016, Batth et al. reported RON localization in the nucleus of DU145 and C4-2B prostate cancer cell lines [55]. They showed that RON behaves as a transcriptional regulator of the AR target gene c-FLIP by binding to its promoter region. This regulation allows cells to adapt to the stress generated by androgen deprivation. Dr. Chang’s group reported that under serum starvation of bladder cancer cells, RON–EGFR complexes translocate to the nucleus where they promote expression of specific target genes belonging to stress response networks. Further work from this group led to the discovery that in bladder cancer cells, nuclear RON interacts with Ku70 and DNA-PKcs to activate NHEJ whereby RON plays a role in hypoxia-induced chemoresistance (Figure 3) [76,77]. The association of RON and DNA-PKcs/Ku70 also occurs when cells growing in hypoxic conditions are treated with doxorubicin or epirubicin. This finding raises the possibility that drugs inducing double-strand breaks may be ineffective in patients with RON overexpression. Similarly, DSB repair was reduced upon EGFR inhibition by gefitinib, erlotinib, or cetuximab [75]. Under hypoxic conditions, activated RON binds to HIF1a and translocates to the nucleus of gastric cancer cells where it can activate c-JUN transcription directly at the promoter locus. In turn, c-JUN promotes cell proliferation and migration (Figure 3) [78]. Recently another group has shown that hypoxia also leads to the binding of HIF1α to RON/RONΔ160–β-catenin complexes; this binding increases nuclear translocation and leads to the expression of transcriptional targets of β-catenin which is a downstream effector of RON (Figure 3). RON and its truncated variant RONΔ160 are both overexpressed in gastric cancer, and RONΔ160 has been shown to promote the growth and migration of gastric cancer cells [78]. The authors showed that binding of HIF1α and RON/β-catenin complexes are essential for gastric cancer cells to adapt to hypoxic conditions and acquire metastatic phenotypes. Similar observations regarding the role of β-catenin in RON induced tumorigenesis were reported in breast cancer [79].

Nuclear localization and subcellular localization of RTKs should be taken into consideration when looking for treatment options. Indeed, nuclear proteins remain accessible to small molecule inhibitors but cannot be targeted by antibodies or targeting peptides. While RON can be translocated jointly with EGFR, one can speculate that other partner RTKs could be involved in similar mechanisms and may vary by cancer type. Thus far, limited information is available regarding RON subcellular localization in cancer. Further work is required to better characterize RON localization and function-specific cancers where targeting RON is of interest.

## 9. Clinical Trials

Pre-clinical studies have demonstrated the therapeutic benefit of RON inhibition using monoclonal antibodies which impede extracellular MSP binding or small molecule inhibitors, which competitively inhibit kinase activation. Several researchers have demonstrated that RON inhibition can sensitize tumors and elicit a profound therapeutic response to a secondary agent [18,19,20,80]. Given this, several early phase human clinical trials are evaluating the safety and efficacy of RON inhibition alone and in combination with other treatment drugs across several cancer types. Here, we reviewed trials whereby RON inhibition is specified in the study details (Table 1).

A Phase I, open label, multi-center, dose-escalation clinical trial was conducted from May 2010 to November 2013 and evaluated the safety profile, efficacy, pharmacokinetics, and pharmacodynamics of monoclonal antibody Narnatumab/IMC-RON8/LY3012219 in patients with advanced solid tumors refractory to standard treatment. The antibody targets the ligand binding domain of RON with 8-fold higher affinity than the natural ligand. Thirty-nine patients were treated with escalating IV drug doses from 5 to 15 mg/kg on a weekly basis or 15 to 40 mg/kg on a biweekly schedule. Overall, the drug was well-tolerated with hyperglycemia as the most common grade 3 adverse effect and only one dose limiting toxicity (DLT) consisting of neutropenia. There were no complete or partial responses with 11 of 39 patients demonstrating stable disease. Twenty-one patients had progressive disease within the first two cycles of therapy. However, it is critical to note that only one patient maintained a drug concentration above 140 ug/mL, at which anti-tumor activity occurred in the animal model [81,82]. The program was abandoned due to concerns regarding the inactivity against multiple RON isoforms, particularly short-form RON which lacks the extracellular domains recognized by the antibody.

The Phase I study NCT02207504 focused on the maximum tolerated dose (MTD), associated toxicities, and pharmacokinetic profile of crizotinib, a c-MET/RON small molecule inhibitor alone and in combination with standard dosing of enzalutamide in castration-resistant prostate cancer patients. This combination was guided by pre-clinical data demonstrating increasing expression of c-MET/RON in multi-regimen disease failure. Crizotinib MTD was found to be 250 mg PO bid and dosed with enzalutamide 160 mg qd. The results were notable for a significant reduction in systemic exposure of crizotinib by 74% attributed to increased hepatic CYP3A4 clearance by enzalutamide, rendering dosing subtherapeutic [83]. This made the associated side effects and any possible drug benefits difficult to attribute to c-Met/RON inhibition. Five patients had stable disease for 20–36 months while those previously exposed to enzalutamide had a progression-free survival (PFS) l of 2.8 months. It is unlikely c-Met/RON inhibition contributed to disease modification given the hepatic clearance rate [83,84].

A Phase II, non-randomized, three parallel-arm cohort study examined the efficacy of crizotinib in advanced urothelial cancers that either highly express c-MET, RON, or the combination thereof. It launched in 2016 aiming to measure overall response rate, overall survival, and overall progression-free survival. The Pfizer-sponsored clinical trial ultimately closed due to poor accrual. No results have been published to date [85].

NCT02745769 was a Phase Ia/Ib, multi-center, non-randomized, open label study evaluating the use of ramucirumab, a VEGF inhibitor, with another c-MET/RON inhibitor, Merestinib/LY280165,3, in Stage IV colorectal cancer. The total number of patients was 23. A second treatment arm included abemaciclib, a CDK4/6 inhibitor, as well as evaluation of Mantle Cell Lymphoma. However, the latter two were dropped. Results included safe dual treatment administration and a tolerable side effect profile with 43% of patients experiencing Grade 3 or higher side effects. Overall, there were no partial or complete responses. Stable disease was observed in 52% of patients, mPFS was 3.3 months, and mOS was 8.0 months [86].

Merestinib/LY2801653 was evaluated in NCT01285037, a multi-center, open label, Phase I study aimed at dose recommendation as well as safety and efficacy when used in combination with cetuximab, cisplatin, or gemcitabine. The expansion cohorts looked to evaluate safety and anti-tumor activity against colorectal cancer (CRC), uveal melanoma (UM), HNSCC, cholangiocarcinoma (CCA), and gastric cancer. The treatment groups consisted of merestinib alone, merestinib + cetuximab, merestinib + cisplatin +/− gemcitabine, and merestinib + ramucirumab, respectively. Overall, 186 patients were treated with 59 (32%) achieving a best response of stable disease, 90 (48%) had disease progression, and 1 death occurred secondary to dyspnea as an adverse event. The median PFSs were 1.7 months (HNSCC), 1.8 months (CRC), 1.8 months (UM), and 1.9 months (CCA). Interestingly, only four PR or CR were observed across all cohorts, and all were within the cholangiocarcinoma treatment groups [87]. Three partial responses were observed in the triple therapy cholangiocarcinoma cohort and one complete response in the dual therapy cholangiocarcinoma group [83]. These findings lead to a subsequent Phase 2 randomized clinical trial in patients with advanced or metastatic biliary cancer. It consisted of two experimental arms. The first included VEGF inhibitor ramucirumab/LY3009806 plus cisplatin and gemcitabine versus placebo plus cisplatin and gemcitabine. In the second arm, merestinib was the designated experimental drug. The aims of the trial were to evaluate for overall survival, overall response, as well as the pharmacokinetics of each experimental treatment. The study was completed February 2018 and while merestinib was well-tolerated and did improve overall response rates, it failed to improve overall survival, progression-free survival, or disease control rate as compared to the standard of care chemotherapy [88]. The study end date has since been extended to December 2022.

## 10. Conclusions

The RON receptor tyrosine kinase is highly conserved across animal species and is involved in orchestrating cell signaling pathways influencing oncogenesis, inflammation, and cancer. RON was originally described as a transmembrane receptor localized to tissue-specific macrophages and epithelial cells that when overexpressed modifies intrinsic cell signaling pathways with subsequent changes in tumor cells, immune interactions, and the microenvironment promoting disease phenotype. While its role in normal biology protects the host by curtailing immune responses, in the tumor microenvironment, RON appears to suppress anti-tumor immune responses. Recent work evaluating RON tumor specific isoforms, RTK crosstalk, and nuclear activities add complexity to the role of RON in various cancer types. Nonetheless, pre-clinical work in various animal models and cancer types identified RON as an intriguing candidate for drug development to potentially target epithelial cancer cells at the primary site as well as the metastatic niche. Doing so may enable induction of an anti-tumor immune response which may subsequently be enhanced using combination immunotherapy strategies. Clinical trials targeting RON have been unsuccessful though very few have been conducted so far. New trials likely await further advances in our understanding of RON’s role in specific tumor contexts and perhaps the identification of biomarkers of cancers driven by RON signaling.

## Figures and Tables

**Figure 1 cancers-14-02037-f001:**
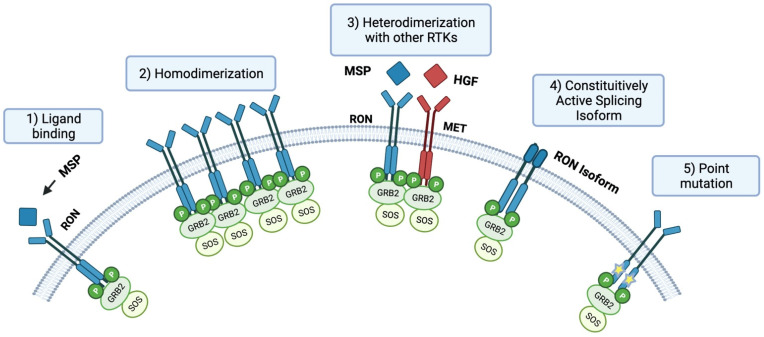
Ron Activation: (**1**) Classical activation via MSP ligand binding induces dimerization and activation. (**2**) Ron overexpression facilitates ligand independent activation via homodimerization. (**3**) Heterodimerization with homolog RTKs including MET. (**4**) Alternative RON isoforms capable of constitutive activation. (**5**) Point mutation RON isoforms capable of constitutive phosphorylation.

**Figure 2 cancers-14-02037-f002:**
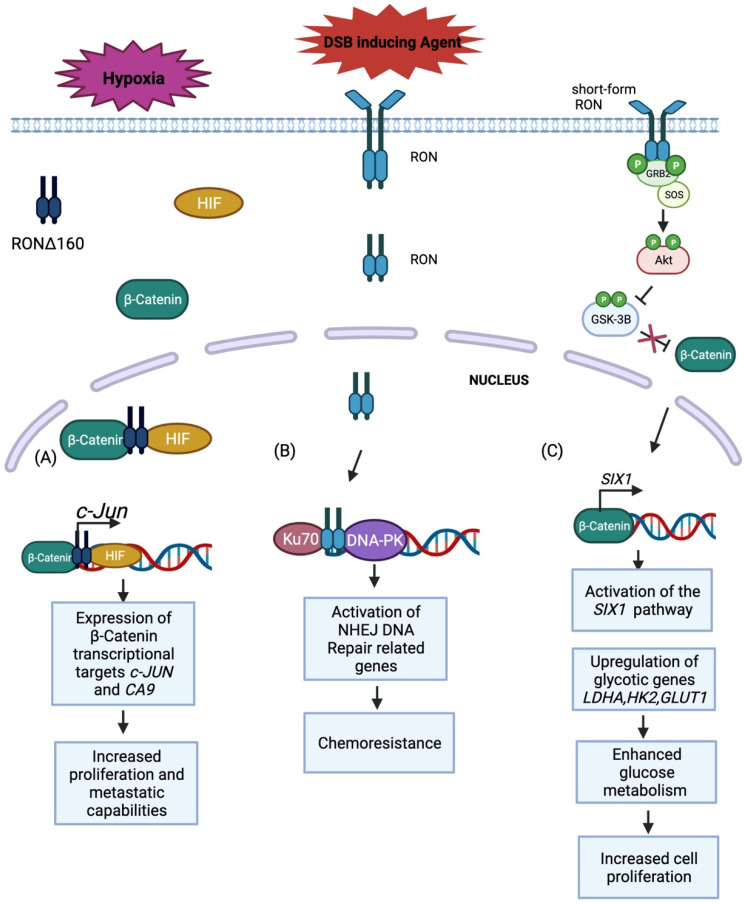
Model demonstrating the role of RON as a transcription factor that promotes cell survival during cellular stress. (**A**) Under hypoxic conditions, a RON splicing variant, RONd160, binds to hypoxia induced factor (HIF) and b-catenin to form a complex that translocates to the nucleus and drives the expression of b-catenin target genes like c-jun and ca-9. The upregulation of these genes leads to increased cell proliferation and metastatic capabilities that drives tumorigenesis. (**B**) In response to treatment with chemotherapeutic agents, RON translocates to the nucleus and binds with Ku70 and DNA-PK to form a complex that drives the expression of genes related to Non-homologous endjoining (NHEJ) pathways. This form of dna repair prevents apoptotic events that would normally be activated due to DSB in DNA, making these cells resistant to chemotherapy. (**C**) A constitutively active form of RON (sf-RON) activates the AKT pathway through phosphorylation. AKT then phosphorylates GSK-3B to inhibit its function. Inactive GSK-3B cannot inhibit B-catenin which is free to enter the nucleus and activate the S1X1 pathway which the drives the expression of glycotic genes. Upregulation of these genes enhances glucose metabolism which increases cell proliferation that is necessary for tumorigenesis.

**Figure 3 cancers-14-02037-f003:**
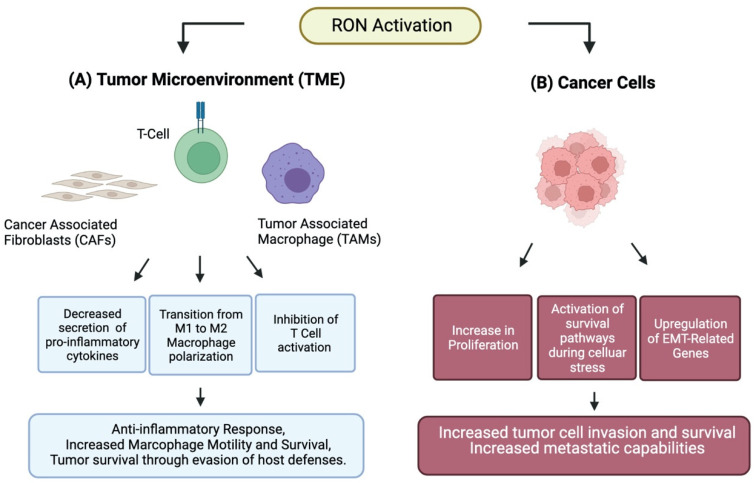
(**A**) RON activation in the tumor microenvironment elicits alteration in cancer associated fibroblasts, T-cells, and tumor associated macrophages culminating in a tumor permissive environment. (**B**) RON activation in cancer cells leads to increased proliferation, activation of survival pathways and increased stress tolerance, as well as upregulation of epithelial to mesenchymal related (EMT) genes.

**Table 1 cancers-14-02037-t001:** Clinical Trials.

Title	Identifier	Sponsor	Phase/End Date	N	Tumor Type	Treatment	Drug Type	Conclusion
A Study of IMC-RON8 in Advanced Solid Tumors	NCT01119456	Eli Lilly and Company	Phase I/Nov 2013	39	Advanced Solid Tumors	IMC-RON8 (Other names) LY3012219 Narnatumab Dose escalation: 5, 10, 15, 20, or 25 mg/kg IV weekly for a 4 week cycle 15, 20, 25, 30, 35, or 40 mg/kg IV every two weeks for a 4 week cycle	Monoclonal Antibody	No complete or partial responses. However, only 1 patient achieved therapeutic drug concentration values >140 μL/mL.
A Phase I Study of LY2801653 in Patients with Advanced Cancer	NCT01285037	Eli Lilly and Company	Phase I/II/Sept 2017	186	Adenocarcinoma of colon or rectum (CRC) HNSCC Uveal Melanoma with liver metastasis (UM) Cholangiocarcinoma (CCA)	Adenocarcinoma: Merestinib HNSCC:Merestinib 120 mg PO × 28 days + Cetuximab UM: Merestinib 120 mg PO × 28 days CCA: Merestinib 120 mg PO qd × 28 days + Cisplatin +/− Gemcitabine Gastric: Merestinib 120 mg PO qd × 28 days + Ramucirumab	LY2801653:small molecule inhibitor c-MET/RON, multi-kinase inhibitor Cetuximab:EGFR inhibitor Cisplatin: alkylating agent Gemcitabine: antimetabolite Ramucirumab: VEGF inhibitor	LY2801653 120 mg qd identified as treatment dose. Three PR and one CR within the triple therapy CCA cohort and dual therapy cohort respectively. Overall, 32% achieved a best response of SD, 48% had PD, and one death occurred due to AE. mPFS 1.7 months HNSCC, 1.8 months CRC, 1.8 months UM, 1.9 CCA, gastric not reported
A Study of Ramucirumab (LY3009806) or Merestinib (LY2801653) in Advanced or Metastatic Biliary Tract Cancer	NCT02711553	Eli Lilly and Company	Phase II/Feb 2018 Modified to end Dec 2022	306	Biliary Tract Cancer Advanced Cancer Metastatic Cancer	A1: Ramucirumab + cisplatin + gemcitabine intravenously (IV) on Days 1 and 8, every 21 days. A2: Placebo + cisplatin + gemcitabine IV on days 1 and 8, every 21 days. B1: Merestinib PO daily + cisplatin + gemcitabine IV on days 1 and 8, every 21 days. B2: Placebo PO daily + cisplatin + gemcitabine IV on days 1 and 8, every 21 days.	Ramucirumab:VEGF inhibitor Merestinib:small molecule inhibitor c-MET/RON, multi-kinase inhibitor	No significant difference in progression-free survival (PFS), overall survival (OS), disease control rate (DCR) between trial drugs and placebo. Secondary endpoint for overall response rate (ORR) noted significant in Merestinib vs. Placebo cohort with two-sided *p*-value 0.015 with Odds ratio 0.4.
LGI-GU-URO-CRI-001: A Phase II Study of Crizotinib in Patients with c-MET or RON Positive Metastatic Urothelial Cancer	NCT02612194	Earle Burgess	Phase II/Nov 2019	46	Stage IV Urinary Bladder Neoplasms Stage IV Ureteral Neoplasms Stage IV Urethral Neoplasms	Cohort 1: Crizotinib c-MET high (>50%) RON null (0–9%) Cohort 2: Crizotinib c-MET (10–100%), RON (10–100%) Cohort 3: Crizotinib c-MET null (0–10%), RON (10–100%)	Crizotinib: c-MET inhibitor	Study terminated due to low accrual.
An Open-Label, Phase Ia/Ib Study of Ramucirumab in Combination with Other Targeted Agents in Advanced Cancers	NCT02745769	Eli Lilly and Company	Phase Ia/Ib/Jan 2019	23	Stage IV Colon Cancer Mantle Cell Lymphoma	Arm 1: Ramucirumab 8 mg/kg IV day 1 and day 15 + Merestinib 80 mg PO qd × 28 days until disease progression. Arm 2: Ramucirumab 8 mg/kg IV day 1 and day 15 + Abemaciclib PO bid × 28 days until disease progression. (Cancelled without enrollement)	Ramucirumab:VEGF inhibitor Merestinib:small molecule inhibitor c-MET/RON, multi-kinase inhibitor Abemaciclib: CDK4/6 inhibitor	Therapeutic doses achieved in combination. In mCRC 43% of patients with >Grade 3 adverse effects with dual treatment. Stable disease was noted in (52%), partial response (0%), complete response (0%). mPFS was 3.3 months. mOS was 8.0 months.
A Phase I Study of Crizotinib in Combination with Enzalutamide in Metastatic Castration-resistant Prostate Cancer Before or After Progression of Docetaxel	NCT02207504	Dana-Farber Cancer Institute Pfizer Astellas Pharma Inc	Phase I/Jan 2022	24	Castration Resistant Prostate Cancer	Crizotinib PO qd 250 mg qd + Enzalutamide 160 mg qd Crizotinib PO 200 mg bid+ Enzalutamide 160 mg qd Crizotinib 250 mg bid + Enzalutamide 150 mg qd	Crizotinib: small molecule inhibitor of c-MET/RON, ROS, ALK Enzalutamide: androgen receptor inhibitor	Crizotinib 250 mg bid was identified as the maximum tolerated dose Concurrent treatment with Crizotinib and Enzalutamide resulted in a significant 74% reduction in systemic Crizotinib exposure likely attributed to enzalutamide inducing CYP3A4.

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
