# Peer review of "The MST1R/RON Tyrosine Kinase in Cancer: Oncogenic Functions and Therapeutic Strategies"

_cancers, 2022, doi:10.3390/cancers14082037_

Round 1

Reviewer 1 Report

The authors well summarized the RON receptor, only need some minors to adjust before being accepted.

Line 41, authors need to represent relevant references of interacting with different pathways by RON.

Line45, the CD34+, the plus should be superscript, and also need apply to the rest of the draft.

Line 87, IFN-g, using the right format.

What about the changing of inflammatory cytokines by RON, such as IL1-beta, IL6?

Line 118. arginase I, A need be capital.

Lin151, the relevant references are not correct.

Line167, what is PanIN.

Line272, missing the reference.

The data of the clinical trial didn't give too much encouragement.

Author Response

File attached 

Reviewer 2 Report

This review is on an intersting topic and is quite comprehensive. However, it was not easily read, at least not by me. In my opinion, the authors need to better structure the manuscript. Start with general background on the protein(s): what is the function in healthy tissue (now only in l.99), which pathways are involved;  subsequently contiunue on the role in diseased tissue.

Similarly, the authors have inlcuded a lot of evidence from literature. In several instances, these experiments need a proper introduction: why are they introduced here? Sometimes this becomes nly clear at the end of the section. Sometimes authors forget to mention whether the experiments were performed in mice, cell lines or humans; or mention 30%, but not of how many patients. Sometimes on the other hand a lot of details from such mannuscripts are included, decreasing readability, and not always leading to clearly explained conlcusions. Sometimes less is more.

They also need to make sure that the first time that an abbreviation is used it is written in full. An English language is quite good, but needs some attention still (hyphens, commas, etc).

Figures are illustrative and informative, legends need some attention.

Overall, very interesting, a lot of work has been performed, and after rewriting/restructuring it, it is certainly worth publication. It will be helpful when the authors start the manuscript with an overview of what is to be expected for the reader.

Author Response

FIle attached

Round 2

Reviewer 2 Report

The authors have signififcantly improved their manuscript. It is now well written, clear, and pleasant to read. The subheadings are more clear, and various sections are better introduced, and contain clear conclusions.

Author Response

The minor edits have been incorporated
